# Current and future levels of mercury atmospheric pollution on global scale

Jozef M. Pacyna [1,6], Oleg Travnikov [2], Francesco De Simone [3], Ian M. Hedgecock [3], Kyrre Sundseth [1], Elisabeth G. Pacyna [1], Frits Steenhuisen [4], Nicola Pirrone [5], John Munthe [7] and Karin Kindbom [7]

[1] Department example, University example, city, postal code, country
[2] Laboratory example, city, postal code, country

[1] NILU – Norwegian Institute for Air Research, Instituttveien 18, PO Box 100, NO-2027 Kjeller, Norway

[2] Meteorological Synthesizing Centre - East of EMEP, 2nd Roshchinsky proezd, 8/5, office 207, 115419 Moscow, Russian Federation

[3] CNR – Institute of Atmospheric Pollution Research, Division of Rende, UNICAL-Polifunzionalie, Rende 87036, Italy

[4] University of Groningen, Arctic Centre, Aweg 30, 9718 CW Groningen, The Netherlands

[5] CNR – Institute of Atmospheric Pollution Research, Area della Ricerca Roma 1, Via Salaria Km 29,300, 00015 Monterotondo (Rome), Italy

[6] Gdansk University of Technology (GUT), Chemical Faculty, 11/12 G, Narutowicza Str., PL-80-952 Gdansk Wrzeszcz, 15  Poland

[7] IVL  Swedish Environmental Research Institute, PO Box 53201, 400 14 Gothenburg, Sweden

*Correspondence to*: Jozef M. Pacyna (jp@nilu.no)

**Abstract.** An assessment of current and future emissions, air concentrations and atmospheric deposition of mercury
worldwide are presented on the basis of results obtained during the performance of the EU GMOS (Global Mercury Observation System) project. Emission estimates for mercury were prepared with the main goal of applying them in models to assess current (2013) and future (2035) air concentrations and atmospheric deposition of this contaminant. The artisanal and small-scale gold mining, as well as combustion of fossil fuels (mainly coal) for energy and heat production in power plants and in industrial and residential boilers are the major anthropogenic sources of Hg emissions to the atmosphere at
present. These sources account for about 37% and 25% of the total anthropogenic Hg emissions globally, estimated to be about 2000 tonnes. The emissions in Asian countries, particularly in China and India dominate the total emissions of Hg. The current estimate of mercury emissions from natural processes (primary mercury emissions and re-emissions), including mercury depletion events, were estimated to be 5207 tonnes per year which represent nearly 70% of the global mercury emission budget. Oceans are the most important sources (36%) followed by biomass burning (9%). A comparison of the
2035 anthropogenic emissions estimated for 3 different scenarios with current anthropogenic emissions indicates a reduction of these emissions in 2035 up to 85 % for the best case scenario.

Two global chemical transport models (GLEMOS and ECHMERIT) have been used for the evaluation of future mercury pollution levels considering future emission scenarios. Projections of future changes in mercury deposition on a global scale

simulated by these models for three anthropogenic emissions scenarios of 2035 indicate a decrease of up to 50 % deposition in the Northern Hemisphere and up to 35 % in Southern Hemisphere for the best case scenario.

The EU GMOS project has proved to be a very important research instrument for supporting, first the scientific justification for the Minamata Convention, and then monitoring of the implementation of targets of this Convention, as well as, the EU Mercury Strategy. This project provided the state-of-the art with regard to the development of the latest emission inventories for mercury, future emission scenarios, dispersion modelling of atmospheric mercury on global and regional scale, and source – receptor techniques for mercury emission apportionment on a global scale.

## 1 Introduction

Mercury (Hg) has been recognized as a toxic, persistent, and mobile contaminant. This contaminant does not degrade in the environment and it is mobile because of the volatility of the element and several of its compounds. It has the ability to be transported within air masses over very long distances. High doses of organic compounds of mercury, particularly methylmercury (MeHg,) can be fatal to humans but even relatively low doses can seriously affect the human nervous system. Mercury has also been linked with possible harmful effects on the cardiovascular, immune and reproductive systems. Methylmercury passes through both the placenta and the blood-brain barrier, so exposure of women of childbearing age and of children to methylmercury is of greatest concern. Consequently, several studies have been conducted on the behavior of mercury in the environment (e.g. Pirrone and Mason, 2009; AMAP/UNEP 2013) and its environmental (e.g. Lindberg et al 2002; Sunderland and Mason, 2007); Mason, 2009; Pacyna et al., 2010; Amos et al., 2012; Lei et al., 2013, Driscoll et al. 2013; Chen et al., 2014; Song et al., 2015, Cohen et al., 2016; Gustin et al., 2016), human health (e.g. AMAP, 2009, Karagas et al., 2012; Sundseth et al., 2015) and economic consequences (e.g.:Ambio, 2007; Sundseth et al., 2010).

The major conclusion drawn from recent studies on the impacts of mercury on the environment and human health is that there is a need for international action to reduce emissions of and human exposure to mercury on a regional and global scale. The EU Mercury Strategy was launched in 2005 (and reviewed in 2010) to support and encourage European-wide action on mercury reduction and to ban its use. The EU's Mercury Strategy provided a comprehensive plan incorporating actions addressing mercury pollution both in the EU and globally. It identified a variety of actions to decrease mercury emissions, cut supply, reduce demand and protect against exposure, especially to methylmercury found in fish. An overview of EU regulations and directives on mercury emissions can be found in Sundseth (2012).

In 2013 under the United Nations Environment Programme (UNEP), countries signed the Minamata Convention on Mercury, a legally binding agreement intended "to protect human health and the environment from anthropogenic emissions and releases of mercury and mercury compounds" (Article 1 in UNEP 2013). The Convention builds upon scientific knowledge of global sources and supply, sinks, and reservoirs of mercury, coupled to linkages with human and wildlife exposures and related health impacts.

Implementation of targets of the Minamata Convention, the EU Mercury Strategy and other policies aiming at the reduction of mercury emissions and their impacts requires an accurate assessment of mercury behavior in the environment. It became clear that the atmosphere is the major transport pathway for the global distribution of mercury. A part of the emissions entering the atmosphere is locally deposited to aquatic and terrestrial ecosystems. Another part is transported with air masses in directions dependent on many factors, including wind direction and speed, and mercury behavior during this transport. As a consequence, mercury emitted in one part of the world can be transported to another. However, the spatial distribution of mercury concentrations and deposition is quite uneven. The question arises from decision makers whether analytical tools are now available to accurately assess these source- receptor relationships for mercury. The Global Mercury Observation System (GMOS) project (www.gmos.eu) has undertaken studies addressing this question. Both, monitoring and model simulations were used for this purpose. The main results from the assessment of current and future emissions, air concentrations and atmospheric deposition of mercury worldwide are presented in this paper.

## 2 Assessment of emissions and future emission scenarios

Emission estimates for mercury were prepared with the main goal of applying them in models to assess current (2013) and future (2035) air concentrations and atmospheric deposition of this contaminant.

### 2.1 Methodology

The approach to estimate the current and future mercury emissions consisted of three steps; i) compilation of the current and future activity data, such as data on consumption of fuels and raw materials and production of industrial goods, ii) link these activities to a compilation of unabated emission factors (UEFs) to derive estimates on unabated emissions to air, and iii) characterization of the effectiveness of air pollution control devices (APCDs) or waste practices and their current and future degree of application. The conceptual approach used to produce the scenario inventories is based on the methodology developed in AMAP/UNEP (2013), illustrated in Annex A. Detailed descriptions of all data sources, calculation methodology and results for 2010, including a comparison with results from national official emission inventories and other publications, can be found in AMAP/UNEP (2013). The methodology for estimating future mercury emissions is consistent with the methodology developed in AMAP/UNEP (2013) for the year 2010. It includes the development of two database modules, one on the projected future activities and the other one on emission factors and emission reduction technology employed in the future for different countries. In this way the emission changes between 2010 and 2035 can be analyzed.

### 2.2 Database on activities

Sector activities relate to national statistics on consumption or production of industrial raw materials or outputs for each mercury- emitting economic sector. Current (2010) statistical data presented in AMAP/UNEP (2013) were collected from national and international experts, international organizations (such as UNEP and International Energy Agency (IEA)),

industry associations and national bureaus (such as the US Geological Survey - USGS). These current statistics were then linked to future projections, supported by several official database sources, such as the IEA, the International Monetary Fund (IMF), the World Bank (WB), the Organisation for Economic Co- operation and Development (OECD) and the United Nations (UN). Future activity data for the year 2035 were estimated and compiled from three data sources/ methodologies in the following way:

- to estimate future energy consumption and production data, the UNEP 2010 estimates were projected in line with the IEA projections presented in the World Energy Outlook (WEO) 2012,

- to estimate the various country- specific industrial goods consumption and production data in the future, a methodology consisting of a year 2035 forecast was developed based on a simple regression model that relates industrial production to a nation`s Gross Domestic Product (GDP) per capita (representing the per capita market value of all goods and services produced within a country). The model fitted a straight line through the set of points for all countries with the resulting slope representing the correlation between national GDP per capita PPP (purchasing power parity) and national annual production of industrial goods. The future projection was then estimated on the basis of forecasting industrial production on the expectations of development of GDP per capita PPP in various countries, based on the OECD database on previous and current GDP per capita PPP as well as the IMF future expectations on GDP per capita PPP,

- to estimate the intentional use of mercury in products, an assumption on voluntary future reductions was made. Assumptions on various degrees of reduced use of mercury in products were based on previously observed trends for different regions (AMAP, 2010) in combination with expectations on implementation of the Minamata Convention. Regional consumption figures used as the basis for the scenarios are presented in UNEP/AMAP (2013), distributed between countries in each region, based on GDP PPP comparisons.

### 2.3 Database on emission factors and future emission reduction technology employed

Country- specific unabated emission factors based on expert evaluation and national data were compiled for AMAP/UNEP (2013). Furthermore, using the method developed in AMAP/UNEP (2013), countries have been assigned to 5 groupings representing different levels of technological implementation (technological profiles) of APCDs. These technologies were characterized by their effectiveness of emission control and degree of application in a given industrial technology. Various assumptions on future application were then made by assuming various step-by-step technology improvements for each country compared to the 2010 situation. The technological profiles were then applied to the 2010 uncontrolled emission estimates and the future activities for the countries/ sectors, resulting in national sector-estimates of unintentional mercury emissions to the atmosphere.

Mercury emissions to air from wastes associated with sectors using mercury intentionally for the various countries were based on world region consumption data as well as on assumptions regarding rates of breakage, degrees of waste handling/incineration and suitable emission factors. The consumption data were distributed between the countries based on

GDP per capita PPPs, see Annex 3 and 4 in UNEP/AMAP (2013). Four different categories of waste management practices (such as waste recycling, controlled- or uncontrolled incineration and landfilling) were assigned to individual countries, based on GDP PPP, where group 1 is the most advanced while group 4 has the least developed practices. Various assumptions on future projections on consumption, as well as the waste management practices and emission factors

constituted the emission scenario from sectors using mercury intentionally. Emissions from use of mercury in Artisanal Gold Mining was estimated based on consumption patterns and assumed emissions to air from different methods employed in different regions, see Annex 2 in UNEP/AMAP (2013).

## 2.4 Definition of emission scenarios

Three main sets of projections were chosen as the basis for compiling future (2035) mercury emissions:

*The Current Policies Scenario (CPS):* The scenario assumes that governmental policies and measures existing in 2010 are adopted, including those that have not been fully implemented. This includes the implementation of traditional APCDs, but also those measures designed to prevent climate change as well as address other environmental problems through energy efficiency and switching to lower carbon fuels. The WEO CPS for 2035 was adopted for the energy sector. The scenario does not include likely but yet undecided future policy initiatives. Thus, it does not forecast the future situation, but it gives

rather a baseline vision on energy, industrial goods and products consumption and production as well as the use of APCDs and waste management practices that are likely to change given no additional effort with regard to policymaking.

*The New Policies Scenario (NPS):* The scenario assumes that policy commitments and plans announced by countries worldwide to reduce greenhouse gas (GHG) emissions, as well as phase out fossil- energy subsidies, are fully implemented. National climate commitments relate to the period of 2020, but additional measures are assumed to be implemented at the

2010 to 2020 pace for the period 2020 to 2035. Future consumption/production of industrial goods is assumed to be at the same level as in the CPS, while the use of mercury in products is assumed to be reduced by 70% in 2035 compared to the 2010 situation as a result of agreements within the Minamata Convention. It is furthermore assumed that all countries will move one step up into more advanced waste practices compared to 2010.

*The 450 [ppm] Scenario (450ppm):* The scenario sets out a target of all counties reaching the highest feasible/available

reduction efficiency in each emission sector. The scenario is not a very realistic one, but it illustrates the maximum possible mercury emission reductions that could be achieved if no other constraints are taken into account, such as economy and increased demand. It can be seen as a "green scenario" that is aiming for a maximum reduction of negative externalities. In the energy sector, it is consistent with a 50% chance of limiting the average global temperature to 2 degree C (compared to pre- industrial levels). This requires that the concentrations of greenhouse gases in the atmosphere are 450 ppm of carbon

dioxide equivalents. The scenario thus features the participation of major economies, such as China and India in the OECD global cap-and –trade scheme after 2020.

A ratio similar to the difference in the IPCC A1 [1] and IPCC B1 [2] scenario were applied to estimate the future consumption/production of industrial goods.

Consumption of mercury in mercury- added products are assumed to be lowered by 95% in 2035 compared to the average in 2010 and a highest possible combination of measures are being applied by all countries which includes collection and safe storage of 15% of mercury in mercury- added products, recycling of 45% of mercury in the waste stream, a lower emission factor (0.03) for controlled waste incineration, assuming at the same time that 100% of waste incineration is applied and that 80% of waste to landfills are safely controlled. Similarly use of mercury in Artisanal Gold Mining was assumed to be reduced by 46% and 76% for the NPS and 450 ppm scenarios, respectively.

### 2.5 Assessment of 2010 global emissions and emission factors

The recent estimate of mercury emissions to the atmosphere (targeting the year 2010) has found artisanal and small- scale gold mining as well as combustion of fossil fuels (mainly coal) for energy and heat production in power plants and in industrial and residential boilers as the major anthropogenic sources of mercury emissions to the atmosphere. These sources account for about 37% and 25% of the total mercury emissions globally, estimated to be about 2000 tonnes. Next, primary non- ferrous metals production and cement production account for relatively large contributions to the emission inventory, being responsible for about 10% and 9 % respectively. Large- scale gold production and waste from consumer products (mostly landfill but also incineration) both contribute about 5% while contaminated sites are responsible for about 4%. Pig iron production contributes about 2.3% while the rest result from the chlor- alkali industry (1.4%), oil refining (0.8%), mercury production (0.6%), cremation (0.2%) as well as natural gas combustion (AMAP/UNEP, 2013).

It should be noted that emission estimates from large-scale gold production are considered preliminary and have large associated uncertainties (AMAP/ UNEP, 2013) The information needed for emission estimates includes the information on the gold content of ore, mercury content of ore, and amount of ore mined per tonne of gold produced. This information varies considerably both between individual countries and within countries – and over time. Currently available information on the above mentioned factors and details on emission estimates for mercury for this sector is available in AMAP/ UNEP (2013).

The emissions in Asian countries, particularly in China and India dominate the total emissions of mercury. This trend has been observed from 2005 until 2010. In fact, Asian emissions also dominated the global anthropogenic emissions of mercury

---

[1]    The A1 scenario describes the future world of very rapid economic growth and a rapid introduction of new and more efficient technologies. It also assumes a substantial reduction in regional differences in per capita income.

[2]    The B1 scenario assumes more environmental focus a rapid change in economic structures towards a service and information economy which reduces material intensity and the introduction of clean and resource- efficient technologies. It assumes, however, no additional climate initiatives.

in the 1990s, as concluded in Pacyna et al (2010). A mercury emission trend assessment has revealed that after having peaked in the 1970`s, the total anthropogenic mercury emissions to the atmosphere appear to be relative stable between 1990 and 2005 (AMAP, 2010). A decrease in emissions in Europe and North America during the time period has been offset by an increase in Asia. The largest increase in emissions is generally due to an increase in coal burning for power and heat

generation and for industrial purposes. Increased use of air pollution controls, removing mercury as a co- benefit (and some mercury specific removing technologies), have slowed down or even reduced the emissions from the increased energy demand. This is especially the case for Europe and North America, but is also reflected in new coal-fired power plants with state-of-art pollution controls installed in China (AMAP/UNEP 2013).

The above-mentioned emission inventory for mercury from anthropogenic sources is the state-of the art. It has been based on

the 2008 background technical report on mercury sources, emissions and transport, summarized by Pacyna et al. (2010).

Spatial distribution of the global anthropogenic emissions of mercury in 2010 is presented in Fig. 1.

The dataset on current emissions from sources other than anthropogenic includes contribution from primary natural sources and reemission processes of historically deposited mercury over land and sea surfaces. The latter source includes reemission of mercury deposited due to historical emissions from both anthropogenic and natural sources.  The mercury emitted from volcanoes, and geothermal sources pertains to primary natural sources, whereas the re-emission of previously deposited mercury on vegetation, land or water surfaces is primarily related to land use changes, biomass burning, meteorological

conditions and exchange mechanisms of gaseous mercury at air-water/top soil/snow-ice pack interfaces.

The current estimate of mercury emissions from primary natural mercury emissions and re-emissions, including mercury depletion events, were estimated in the GMOS project to be 5207 tonnes per year which represent nearly 70% of the global mercury emission budget. This emission estimate compares fairly well with the information provided in the latest work by Cohen et al. (2016). The total emission of mercury from biomass burning, geogenic processes, and soil/ vegetation/ ocean re-

emissions was assessed to 6500 tonnes per year, adopted from Lei et al. (2014). Cohen et al. (2016) have reviewed the latest work by Selin et al., (2008), Amos et al. (2012); Chen et al. (2014), Song et al. (2015) and others.

Oceans are the most important sources from natural and reemission sources assessed within the GMOS project contributing of 36% to the emissions of mercury, followed by biomass burning (9%), deserts, metalliferous and non-vegetated zones (7%), tundra and grassland (6%), forests (5%) and evasion after mercury depletion events (3%). Overall, the relative

contribution of terrestrial surfaces is 2429 tonnes per year  and that from surface waters is 2778 tonnes per year . Another estimates of current annual re-emissions to the atmosphere that are a legacy of historical mercury releases from both anthropogenic and natural sources are in the range 4000–6300 tonnes per year (Mason et al., 2012)

It should be noted that the reemission processes in this paper are assumed to be static one-way upward flux. This makes a simplification of global biogeochemical cycle of mercury and as such, it is a limitation of the current analysis.

## 2.6 Assessment of global mercury emissions in the year 2035

Many variables affect future mercury emissions, however, the main ones are likely to be linked to the production and consumption of energy and industrial goods, intentional use of mercury in products, artisanal and small-scale gold mining, the use of dental amalgam, as well as increasing human population and related demands on one side, and the introduction of legislations and directives, awareness campaigns, industrial technology improvements and the increasing use of pollution control equipment on the other side.

Implementation of various climate change mitigation options to reduce carbon dioxide emissions, such as improvement of energy efficiency in power stations, replacement of fossil fuels by renewable sources, improvement of combustion, and industrial technologies, and application of carbon capture and storage (CCS) technologies are expected to have positive effects on the reduction of releases of mercury since these measures typically reduce the emissions of mercury as well as several other contaminants of concern for the environment and human health as a co- benefit. A wide range of policies are already in place (mainly in OECD countries) to encourage reduction of greenhouse gas emissions. It is expected, however, that the use of cheaper fossil fuels is likely to remain dominant in most regions to meet increasing energy demands.

Changes in energy production and consumption until the years 2035 and 2050 have been presented by the IEA WEO and Energy Technology Perspectives (ETP), respectively. In the projections, WEO focuses on certain key aspects, such as energy prices, concerns for greenhouse gas emissions and its impacts on energy investments, the increasing use of renewable energy, changes in regulations and directives as well as recent developments in technologies for energy production.

IEA projects that the population is for all the scenarios assumed to expand from 6.7 billion (in 2008) to 8.5 billion in 2035 in which population in non- OECD countries continues to grow most rapidly. In the same period, the GDP is assumed to grow worldwide by 3.2 % per year on average. India, China and Middle East are assumed to grow most rapidly in terms of GDP as well as increase in energy demand. OECD projects that in the next 20-50 years, China will become the world's largest economy whilst India will surpass Japan and catch up with the Euro area before 2030. On average the GDP per capita ppp growth will be roughly 3% annually in the non- OECD area against 1,7% in the OECD area. IEA projects that in the NP (New Policy) scenario, energy demand continues to increase by 40% from 2008 until 2035. In the same period, the energy demand will be about 8 % higher in the CP (Current Policy) scenario and 11 % lower in the 450 ppm scenario in comparison to the CP scenario. Coal remains the dominant energy source in the NP scenario, but the share declines by 7 % in the period to 2035. Coal demand increases by about 25 % (mostly up to 2020), while electricity demand increases by 80 % by 2035. Coal use is assumed to be critically influenced by government policies related to climate change. No change in government policies, strong global economic growth and increased energy demand in non- OECD countries increases global fossil fuels demand substantially in the CP scenario. In contrast, implementation of measures to meet climate targets and policies reduce e.g. coal demand by a quarter in the New Policies Scenario and more than half in the 450 ppm scenario in comparison to the CP scenario. Less coal use is seen for the OECD countries in all scenarios between 2010 and 2035 (WEO, 2011). An

illustration of the assumed coal use until 2035 under the WEO CP, NP and 450 scenario assumptions, is presented in Annex A.

The Intergovernmental Panel on Climate Change (IPCC) Special Report on Emission Scenarios (SRES) (Baseline scenario A1, A2, B1 and B2) on climate change does not project any additional policies above current ones until year 2100, however they focus on socio- economic, demographic and technological change.

An overview of the assumed future consumption of mercury-containing products is available in table A1, table A2, and table A3.

A comparison of the 2035 emissions estimated for various scenarios indicates that 1960, 1020 and 300 tonnes of annual mercury emissions would be emitted globally in 2035 under the CP, NP and MFR (Maximum Feasible Reduction) scenarios, respectively. This means that if mercury continues to be emitted under the control measures and practices that are decided at present against a background of changing population and economic growth, the 2010 emissions will remain the same in 2035. A full implementation of policy commitments and plans (the basic assumption of the NP scenario), implies a benefit of reducing mercury emissions by up to 940 tonnes per year in 2035 under the assumptions employed in this scenario. A maximum feasible emission reduction of mercury emissions results in 1 660 tonnes less emissions than those emissions envisaged under the CP scenario. The sector- and region-specific contributions to mercury emissions under the various scenarios can be observed in Figs. 2 and 3, respectively.

Fig. 2

Fig. 3

Spatial distributions of emissions within various scenarios in 2035 are presented in Figs 4 a, b, and c for the CP, NP and MFR scenarios, respectively.

Fig. 4 a

Fig. 4 b

Fig. 4 c

## 3 Model evaluation of future scenarios of mercury pollution

Various models were used to estimate current atmospheric concentrations and deposition of mercury world-wide, as well as atmospheric deposition of the contaminant in the future.

### 3.1 Model description

Two global chemical transport models (GLEMOS and ECHMERIT) have been used for the evaluation of future mercury pollution levels considering future emission scenarios.

GLEMOS (Global EMEP Multi-media Modelling System) is a multi-scale chemical transport model developed for the simulation of environmental dispersion and cycling of different chemicals including mercury (Travnikov and Ilyin, 2009). The model simulates atmospheric transport, chemical transformations and deposition of three mercury species (GEM, GOM

and PBM). The atmospheric transport of tracers is driven by meteorological fields generated by the Weather Research and Forecast modelling system (WRF) (Skamarock et al., 2009) fed by the operational analysis data from ECMWF (ECMWF, 2016).The model has a horizontal resolution 1°×1°.Vertically, the model domain reaches 10 hPa and consists of 20 irregular terrain-following sigma layers. The atmospheric chemical scheme includes mercury redox chemical reactions in both the gaseous and aqueous phase (cloud water). Oxidized mercury species (GOM and PBM) are removed from the atmosphere by

wet deposition. All three species interact with the ground contributing to dry deposition.

ECHMERIT is a global on-line chemical transport model, based on the fifth generation global circulation model ECHAM, with a highly flexible chemistry mechanism designed to facilitate the investigation of tropospheric mercury chemistry (Jung et al., 2009, De Simone et al., 2014). ECHMERIT uses T42 horizontal resolution (roughly 2.8° by 2.8° at the equator) and 19 vertical non-equidistant hybrid-sigma levels up to 10 hPa. The model simulates the physical and chemical process of three

mercury species (GEM, GOM and PBM). Monthly biomass burning mercury emissions from the FINNv1 inventory are mapped off-line to the model (De Simone et al., 2015), whereas emissions from oceans are calculated on-line, as discussed in (De Simone et al., 2014). Prompt re-emission of a fixed fraction (20%) of deposited mercury is also applied in the model to account for reduction and evasion processes which govern mercury short-term cycling between the atmosphere and terrestrial reservoirs (Selin et al., 2008). This fraction is increased to 60% for snow-covered land and the ice covered seas.

Mercury removal processes include both dry and wet deposition of reactive species (GOM and PBM). GEM does not contribute to dry deposition. The model was run in reanalysis mode using data from the ERA-INTERIM project (ECMWF). Chemical transformations play important role in dispersion and cycling of mercury in the atmosphere. However, current understanding of the mercury oxidation and reduction chemistry in the atmosphere contains significant uncertainty. The oxidation of GEM by Br halogens is generally accepted as a dominant oxidation pathway in a number of atmospheric

environments including the polar regions, marine boundary layer and the upper troposphere/lower stratosphere (Hedgecock and Pirrone, 2004; Holmes et al., 2009; Lyman and Jaffe, 2011; Obrist et al., 2011; Gratz et al., 2015). However, very little data exists with respect to this mechanism in the global atmosphere (Kos et al., 2013). On the other hand, in spite of

theoretical doubts of viability and significance of GEM oxidation by O3 and OH radical under the atmospheric conditions (Calver and Lindberg, 2005; Hynes et al., 2009), there are ample possibilities of occurrence of complex reactions involving these oxidants in the polluted atmosphere in presence of aerosol particles and secondary reactants (Snider et al., 2008; Cremer et al., 2008; Rutter et al., 2012; Subir et al., 2012; Aria et al., 2015). Other possible reactants that can contribute to

mercury transformation in the atmosphere include Cl2 (Aria et al., 2002), H2O2 (Tokos et al., 1998), HCl (Hall and Bloom, 1993), NO3 (Peleg et al., 2015), etc. However, exact mechanisms, reaction products and relative importance of the reactions are still poorly known. More detailed discussion of uncertainties associated with mercury atmospheric chemistry and its implementation in contemporary chemical transport models can be found in (Lin et al., 2006; Subir et al., 2011; 2012; Gustin et al., 2015; Aria et al., 2015).

A majority of chemical transport models that are used for simulations of mercury dispersion on global and regional scales assume the O3- and/or OH-initiated reactions as the main pathways of GEM oxidation in the free troposphere (Christensen et al., 2004; Travnikov and Ilyin, 2009; Pan et al., 2010; Baker and Bash, 2012; Kos et al., 2013; Gencarelli et al., 2014; De Simone et al., 2015). The Br oxidation mechanism is also often applied as an option (Lei et al., 2013; Dastoor et al., 2015) or as the only oxidation pathway for the whole atmosphere (Holmes et al., 2010; Soerensen et al., 2010; Amos et al., 2012;

Shah et al., 2016). It should be noted that recent comparison studies showed that models with diverse formulations of atmospheric chemistry were able to simulate realistic distributions of GEM air concentration and total mercury deposition on a global scale (Travnikov et al., 2010; AMAP/UNEP, 2013; AMAP/UNEP, 2015). In particular, the model derived source attribution of mercury deposition on a continent scale (Europe, North America, Asia, etc.) agreed within 10-15% among different models. Therefore, taking into account the limited knowledge on the overall redox cycle of mercury in the

atmosphere the standard chemical schemes based on the O3- and OH-initiated reactions were applied in this study, which allow reasonable reproduction of the measured mercury concentration and deposition levels on a global scale. Nevertheless, the two models differ in their treatment of the forms of the oxidation products. In GLEMOS all products of GEM oxidation are treated as PBM, whereas ECHMERIT expects the products to be in the gaseous form (GOM).

Both models used the global mercury anthropogenic emission inventory for 2010 (AMAP/UNEP, 2013) for the current state

and the three emission scenarios for 2035 discussed above in Section 2. However, the models use different estimates of natural and secondary emissions of mercury to the atmosphere. GLEMOS utilized prescribed monthly mean fields of mercury emission fluxes from geogenic and legacy sources (Travnikov and Ilyin, 2009), whereas ECHMERIT applied parameterisation of dynamic air-seawater exchange as a function of ambient parameters but using a constant value of mercury concentration in seawater (De Simone et al., 2014). As a result, the total estimate of mercury global natural and

secondary emissions by ECHMERIT (8600 t/y) is a factor of 2 higher than the total value used in GLEMOS (3995 t/y). The higher emissions in the former model are compensated by higher deposition of GEM in the air-surface exchange process. Therefore the net fluxes of mercury exchange between the atmosphere and the surface are comparable in the models.

Meteorological data for 2013 were used in all simulations to exclude the influence of inter-annual meteorological variability on the analysis results. Each model run consisted of a multi-year spin-up to reach steady-state conditions and a one-year

control simulation for the analysis. It should be noted that the geogenic and legacy sources were assumed to be unchanged during the simulation period (2013-2035) by using static fluxes of natural and secondary emissions in one model and constant mercury concentration in seawater in the other. Thus, the results presented reflect the response of mercury atmospheric deposition to changes in direct anthropogenic emissions and do not take into account the possible feedback of

the ocean and terrestrial reservoirs to these changes. Indeed, application of an atmospheric chemical transport model coupled with a mechanistic model of mercury cycling in soil shows that reductions in anthropogenic mercury emissions will lead to rapid decrease in mercury emissions from soil (Smith-Downey et al., 2010). Besides, Amos et al. (2013) applied a fully coupled biogeochemical model and showed that even if anthropogenic emissions stay unchangeable, mercury deposition will continue to increase due to effect of the legacy of oat anthropogenic emissions accumulated in the ocean. Generally, the

atmosphere responds quickly to the termination of future emissions but long-term changes are sensitive to a number of factors including historical changes of anthropogenic emissions, air-sea exchange, mercury burial in deep ocean and coastal sediments, etc. (Amos et al., 2014; 2015).

### 3.2 Assessment of current air concentrations and atmospheric deposition of mercury

Global distributions of surface GEM concentrations simulated by two global models are shown in Fig. 5. The model results

show similar spatial patterns of mercury concentrations with a pronounced gradient between the Southern and the Northern Hemispheres and elevated concentrations in major industrial regions – East and South Asia, Europe and North America., High concentrations are also seen in some regions of the tropics (the north of South America, Sub-Saharan Africa and Indonesia) due to significant mercury emissions from artisanal and small-scale gold mining. Generally, the simulated results agree satisfactorily with observations shown by circles in the figure, using the same colour palette.

Fig. 5

More detailed model-to-measurements comparison of GEM concentrations as well as mercury in wet deposition is shown in the scatter plots in Fig.6 . As can be seen, the discrepancy between the simulated and observed concentrations of GEM does

not exceed a factor of 1.5. It should be noted that the models also demonstrate acceptable performance simulating wet deposition fluxes (not shown here). However, model-to-model and model-to-observation deviations are somewhat larger in this case due to stronger effect of uncertainties in atmospheric chemistry and some meteorological parameters (e.g. precipitation amount). Thus, the models successfully reproduce the spatial patterns of mercury concentration in air and wet deposition under current conditions.

Fig. 6

A more challenging parameter for model simulations is the total atmospheric deposition (wet and dry). In contrast to wet deposition, the dry deposition of mercury, that describes interaction with the surface is poorly known and sparsely measured to constrain chemical transport models (Agnan et al., 2016; Zhu et al., 2016).The situation becomes even more complicated by the bi-directional character of surface uptake (Zhang et al.,2009; Qureshi et al., 2012; Gustin, 2012; Wang et al., 2014). In particular, some terrestrial environments can act as short-term or long-term sinks for mercury, whereas others will act as sources depending on geology of the soil, enrichment with mercury, plant and litter cover. (Gustin, 2012). Moreover, anthropogenic and natural activities such as mining, landfills, wildfires etc. can significantly alter the air-surface exchange (Aria et al., 2015). Therefore, estimates of mercury dry and, consequently, total deposition of mercury by contemporary models differ significantly (Lin et al., 2006; 2007; Bullock et al., 2008; Travnikov et al., 2010; AMAP/UNEP, 2015).

Fig. 7 shows the global distributions of total mercury deposition flux simulated by GLEMOS and ECHMERIT for the current state. Both models estimate high deposition levels over industrial regions of East and South Asia, Europe and North America. There is also significant deposition in the tropics due to precipitation. However, there are considerable differences between the two models caused by different model formulations. ECHMERIT predicts significantly larger deposition levels in low and temperate latitudes, particularly, over the ocean because of intensive air-water exchange. On the other hand, GLEMOS simulates increased deposition fluxes in the polar regions due to the effect of the atmospheric mercury depletion events (Schroeder et al., 1998; Lindberg et al., 2002; Steffen et al., 2008), which are not taken into account by ECHMERIT. As shown below, the differences between the model estimates of mercury total deposition are largely caused by the contributions of natural and secondary sources (see Section 3.1).

Fig. 7

The majority of human exposure and health risk associated with mercury comes from consumption of marine and freshwater foods (Mahaffey et al., 2004; 2009; Sunderland et al., 2010; AMAP/UNEP, 2013). Direct atmospheric deposition is the dominant pathway of mercury entry to the ocean and freshwater environmental compartments (taking into account watersheds) (Mason et al., 2012; AMAP/UNEP, 2013). Therefore, in the following analysis the focus will be placed on changes of mercury deposition in future scenarios with respect to the current state keeping in mind significant uncertainties of available model estimates of this parameter.

**3.3 Future changes of mercury deposition levels**

Projections of future changes in mercury deposition on a global scale simulated by GLEMOS and EHMERIT for three emissions scenarios of 2035 are illustrated in Fig. 8 . The 'Current Policy' scenario (CP 2035) predicts a considerable decrease (20-30%) of mercury deposition in Europe and North America and strong (up to 50%) increase in South and East Asia (Figs. 8 a and 8 b). In other parts of the Northern Hemisphere no significant changes (±5%) are expected, whereas a slight decrease (5-15%) in deposition is seen in the Southern Hemisphere.

According to the 'New Policy' scenario (NP 2035) a moderate decrease of mercury deposition (20-30%) is predicted over the whole globe except for South Asia (India), where an increase in deposition (10-15%) is expected due to the growth of regional anthropogenic emissions (Figs. 8 c and 8 d).

Model predictions based on the 'Maximum Feasible Reduction' scenario (MFR 2035) demonstrate consistent mercury deposition reduction on a global scale with a somewhat larger decrease in the Northern Hemisphere (35-50%) and a smaller decrease (30-35%) in the Southern Hemisphere (Figs. 8 e and 8 f). Thus, the most significant changes in mercury deposition (both increase and decrease) during the next 20 years for all considered scenarios are expected in the Northern Hemisphere and, in particular, in the largest industrial regions, where the majority of regulated emission sources are located.

Fig. 8

## 4 Source apportionment of mercury deposition

Mercury is known to be a global pollutant as it is transported over long distances in the atmosphere. As has been shown in previous studies (Seigneur et al., 2004; Selin et al., 2008; Travnikov and Ilyin, 2009; Corbitt et al., 2011; Lei et al., 2013; Chen et al., 2014), atmospheric transport from distant sources can make a significant contribution to mercury deposition. Therefore, changes of mercury deposition in a region depend not only on the dynamics of local emissions but also on emission changes in other regions of the globe. The global models have been applied for source apportionment of mercury deposition for both the current state and the future scenarios. The definition of source and receptor regions adopted in the study is shown in Fig. 9. The regions considered include the continents (Europe, North, Central and South America, Africa, Australia), large sub-continents (the Middle East, countries of the Commonwealth of Independent States (CIS), South, East and Southeast Asia) and the Polar Regions.

Fig. 9

The dynamics of mercury deposition between 2013 and 2035 in the various geographical regions is shown in Fig. 10 along with the disintegration of the average deposition flux into direct anthropogenic and natural/legacy components. As mentioned above, both models simulate the highest mercury deposition fluxes and the most significant deposition in South, East and Southeast Asia. Mercury deposition increases or decreases by up to a factor of two in these regions depending on the scenario. All other regions are characterized by either insignificant changes (CP 2035) or moderate deposition reduction (NP 2035 and MFR 2035). The smallest changes are expected in regions remote from significant emissions sources (e.g. the Arctic and Antarctica). Deposition fluxes simulated by ECHMERIT are higher by a factor of 1.5-3 than those simulated by GLEMOS. The difference is the greatest over the oceans. It should be noted that the deviation is largely caused by

differences in deposition from natural and legacy sources. Levels of mercury deposition from direct anthropogenic sources simulated by the two models are comparable. Therefore, the following source apportionment analysis was performed for the anthropogenic component of mercury deposition and presented using the mean value of the two models.

Fig. 11 shows the source apportionment of mercury deposition from direct anthropogenic sources in different geographical regions. The contribution of unchanged natural and legacy emissions is not shown in the figure. Mercury deposition in South and East Asia is largely determined by domestic sources (Figs. 11 a-b). In both regions the CP 2035 scenario predicts a
significant increase in deposition during next 20 years. In contrast, the NP 2035 scenario forecasts increasing deposition in South Asia, but a decrease in East Asia. A strong decrease in deposition would be expected in both regions according to the MFR 2035 scenario. All three future scenarios predict a reduction of mercury deposition in North America and Europe (Figs. 11 c-d). The decrease in mercury deposition from local sources is partly offset by an increase in deposition from Asian sources for CP 2035, whereas the other scenarios predict a reduction of mercury deposition from most anthropogenic
sources. The Arctic is a remote region without significant local emission sources of mercury. Changes of mercury deposition in this region reflect the dynamics of major emission sources in the whole North Hemisphere (Fig. 11 e). In spite of a significant emission reduction in Europe and North America, the Arctic is largely affected by mercury atmospheric transport from East and South Asia. As a result, no significant deposition changes are expected in this region according to CP 2035. However, the other scenarios forecast a net reduction of mercury deposition from direct anthropogenic sources in the Arctic.

In a previous study Corbitt et al. (2011) applied a global atmospheric model coupled to the surface reservoirs to quantify 2050 emissions projections base on four emissions scenarios developed by the Intergovernmental Panel on Climate Change
(IPCC). It was obtained that mercury deposition in 2050 will stay similar to the present-day levels for the best-case scenario but will increase for the other scenarios. The largest increase was predicted in Asia (in China and, particularly, in India) mostly because of increased contribution from domestic emissions. Change of mercury deposition in the United States ranges from 30% increase to 10% decrease depending on applied scenario. Lei et al. (2014) considered similar IPCC based scenarios and found increase of wet deposition by 2050 over the continental US in all the cases.
As it was mentioned above the presented results does not account for possible response of the global biogeochemical reservoirs to future changes of anthropogenic emissions. Continues emissions can lead to further accumulation of mercury in soil and the ocean and shift future deposition change toward increase in a medium-term perspective (Amos et al., 2013). Additionally, climate change can also alter future levels of mercury deposition through changes to vegetation cover and atmospheric oxidants, increased wildfires, enhanced air-seawater exchange, etc.

**5 Concluding remarks**

Monitoring the implementation of international agreements mercury emission reduction and its impacts on the environment and human health would require the improvement of various parameters, including emission inventories, model simulations of atmospheric deposition, and monitoring networks. This is particularly important for the monitoring of implementation of global and regional agreements, such as the Minamata Convention, and the EU Mercury Strategy, respectively.

Currently available global emissions inventories are quite complete and accurate for some anthropogenic sources, such as the energy and industrial sectors (an uncertainty of about 25 %). Much less accurate are the emission inventories for waste incineration and artisanal gold mining and production (perhaps up to a factor of 3 uncertainty). Major improvement of global emission inventories for anthropogenic sources is now expected as national emission inventories are now being carried out in several countries in preparation for the requirements posed by the Minamata Convention.

Future emission projections for mercury emissions from anthropogenic sources are dependent on economic development plans in individual countries, particularly energy production plans. One of the first attempts in developing such scenarios is presented in this work. Reduction of mercury in the future can be achieved as a co-benefit when reducing of emissions of greenhouse gases, as well as, through implementation of mercury-specific controls. The choice of future non-fossil energy sources will have large effects on mercury emissions: biomass combustion will continue to mobilise mercury present in the fuels (even if some of this mercury is natural) whereas non-combustion solutions such as solar or wind based power generation will of course not cause additional emissions of mercury.

Major problems still exist with the development of mercury emission inventories for natural sources and re-emission of this contaminant. Existing emission estimates vary by a factor of 3. This is difficult to accept for assessing current and future levels of atmospheric deposition of mercury. More measurements are needed to improve the accuracy of mercury releases from re-emission of this pollutant from contaminated sites.

The results of this study confirm that current models can adequately simulate transport and atmospheric deposition of mercury. However, the accuracy of these simulations depend on the quality of input parameters , such as emissions data, meteorological parameters and modules describing the chemical and physical behavior of mercury and its compounds after entering the atmosphere. Taking into account the above mentioned concerns about mercury emission inventories, the models in this study could describe properly the atmospheric deposition trends at present and in the future. This has been confirmed by comparison of models estimates and ground-site measurements. It has been shown that the major environmental problem with mercury pollution is and will be in south-east Asia, where current emissions are the by far the largest, compared with emissions in other regions. Although mercury is a global pollutant, it has been shown by measurements and model estimates that the greatest air concentrations and atmospheric deposition is in the regions of the contaminant largest emissions. This is a clear message to policy makers developing plans for reduction of human and environmental exposure to mercury.

At present, reliable source – receptor techniques are available to study the relationship between emissions and atmospheric deposition of mercury on a global scale. This information is particularly important when elaborating strategies for mercury

emission reductions worldwide. As in the case of dispersion models, the quality of estimates using the source − receptor techniques depends on many factors, including the quality of emission data, and the accuracy of measurements and model simulation of mercury atmospheric deposition. This important issue has been discussed recently in Pirrone et al. (2013) Gustin et al. (2016), and Cohen et al (2016).

## 5 6 Acknowledgements

The results presented in this paper have been obtained within the EU Global Mercury Observation System (GMOS) project (www.gmos.eu). The GMOS project has been carried out with financial support from the EC (Grant Agreement No. 265113). The authors are grateful for this support. Financial and other contributions to the development of the methodology for emission inventories and projections from UNEP Chemicals and AMAP are also acknowledged.

10 The authors would also like to thank other scientists from the GMOS project for the possibility of using the measurement data when discussing the outcome of model simulations with monitored data.

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

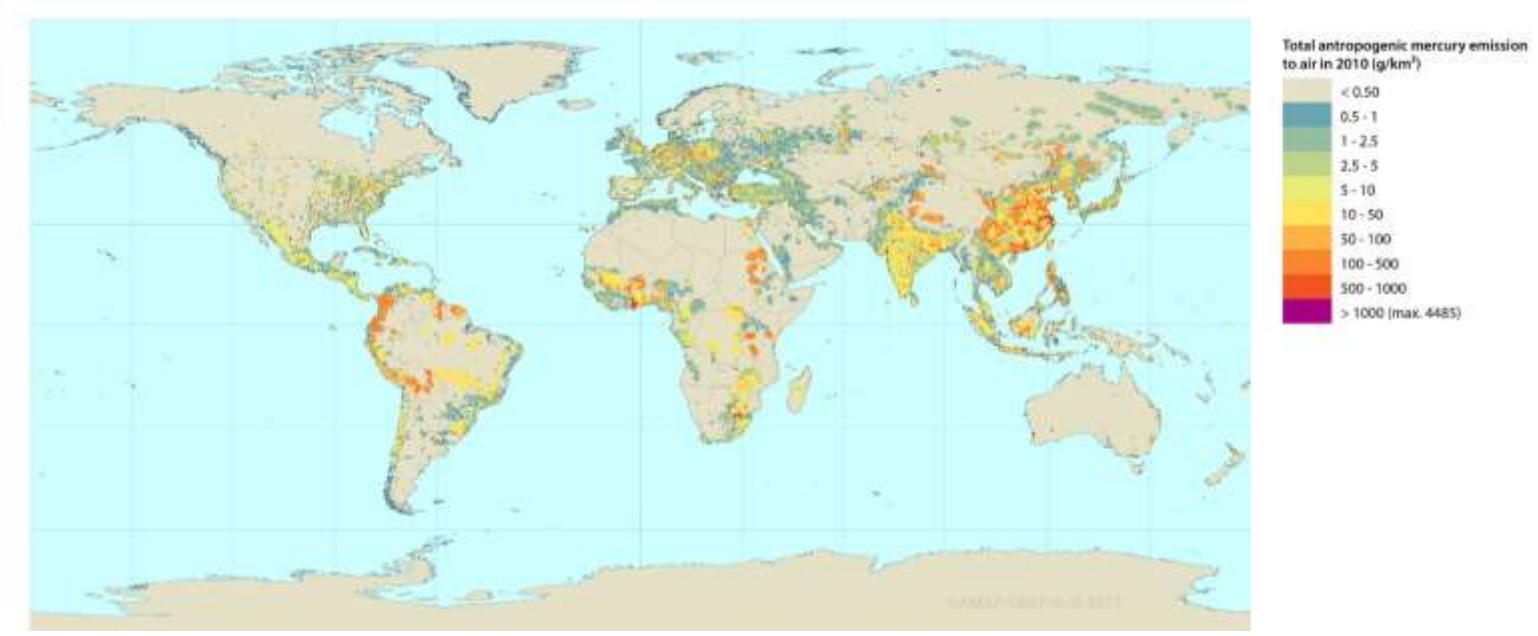

**Figure1. Spatial distribution of global anthropogenic emissions of mercury in 2010.**

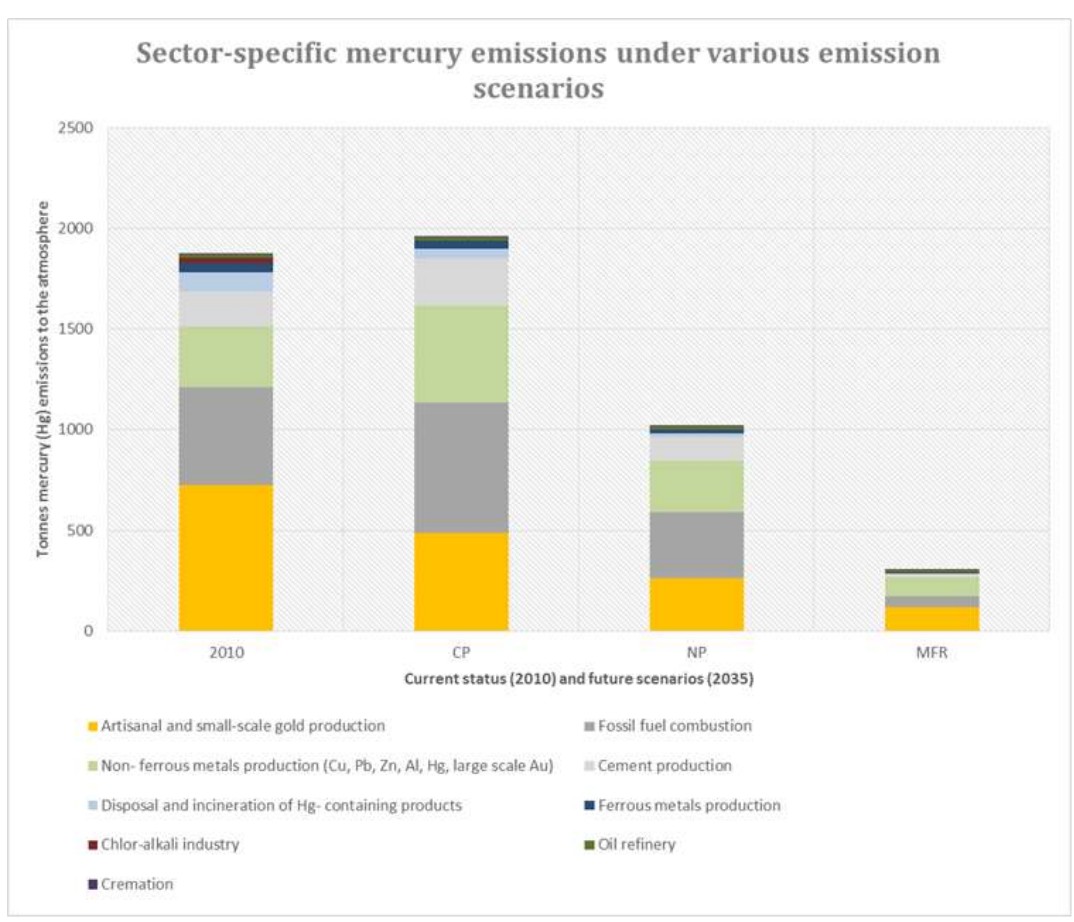

**Figure 2. Sector specific emissions of mercury under various emission scenarios in 2035**

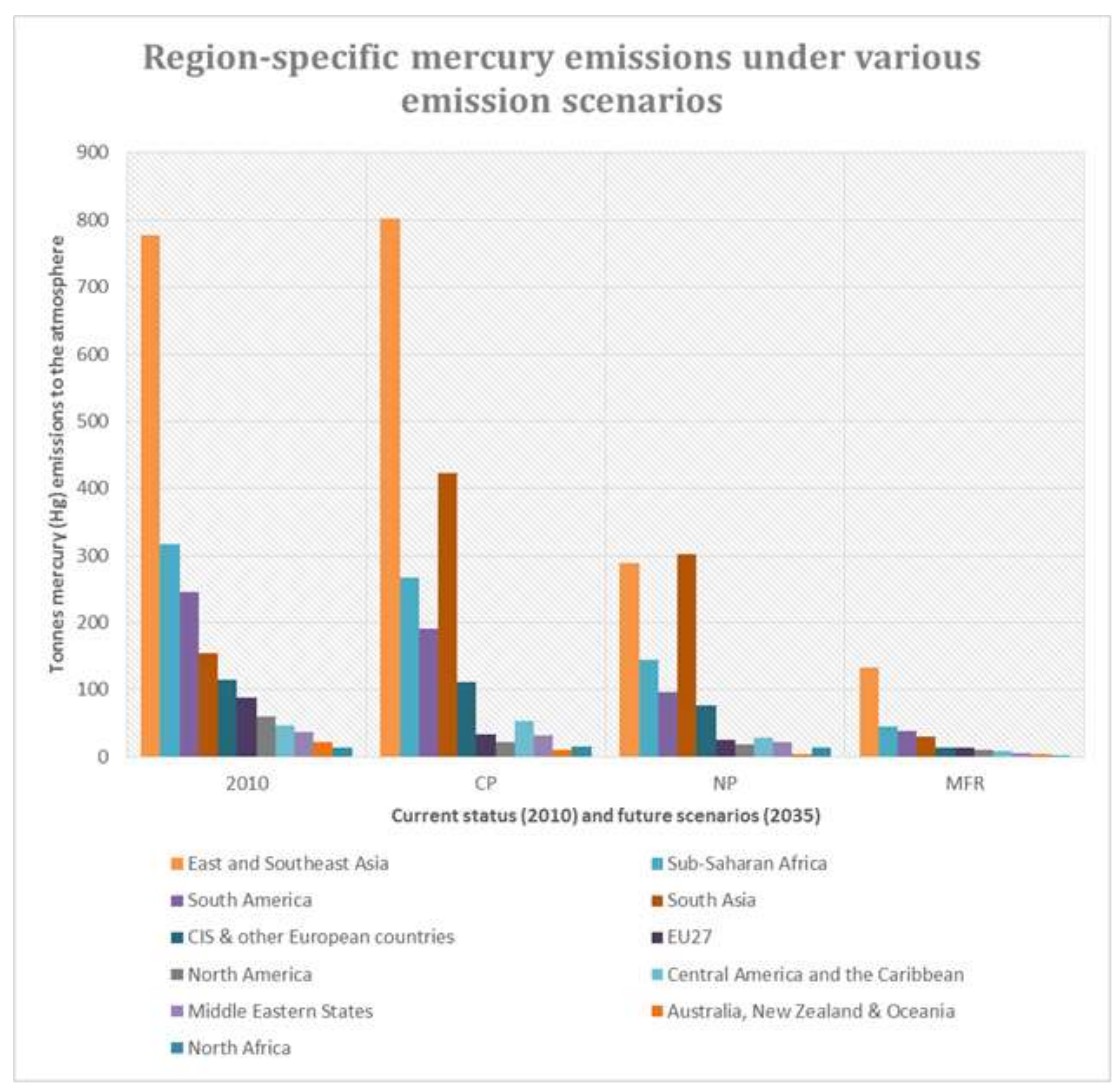

**Figure 3. Region specific emissions of mercury under various emission scenarios in 2035.**

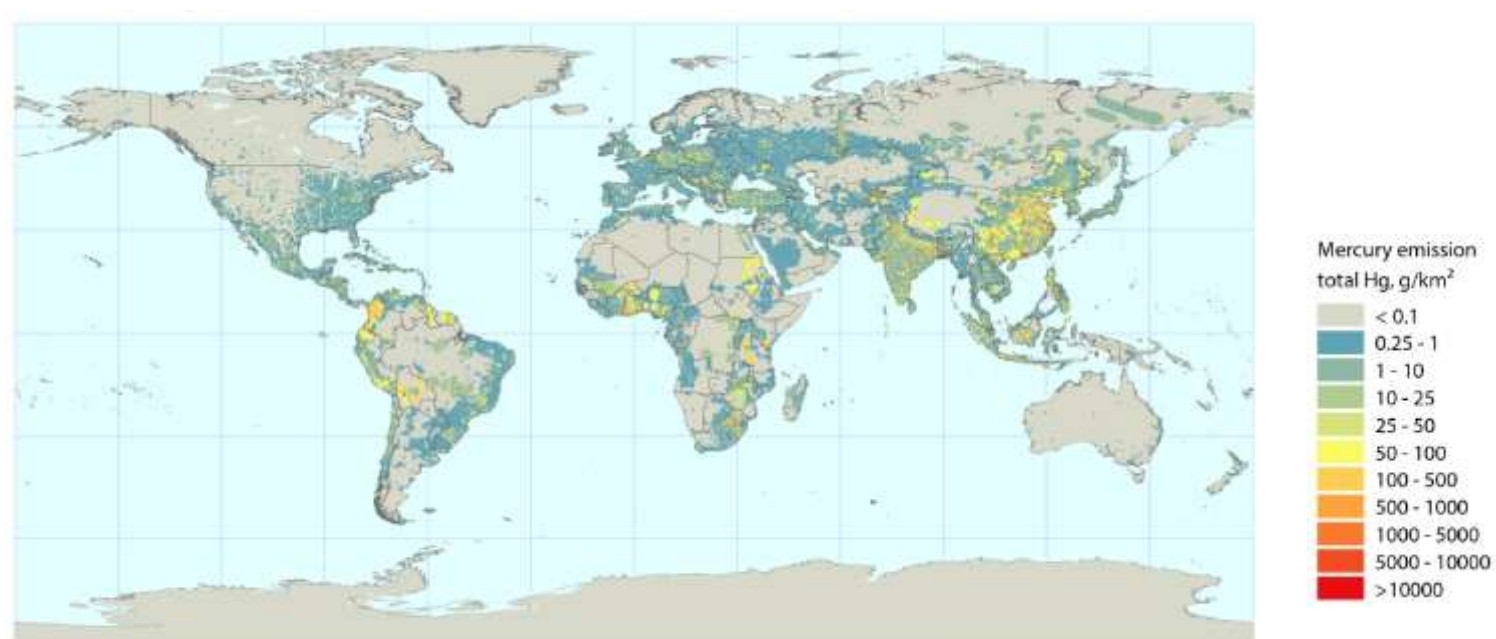

Current policy

**Figure 4 a. Spatial distribution of mercury emissions in 2035 according the CP scenario.**

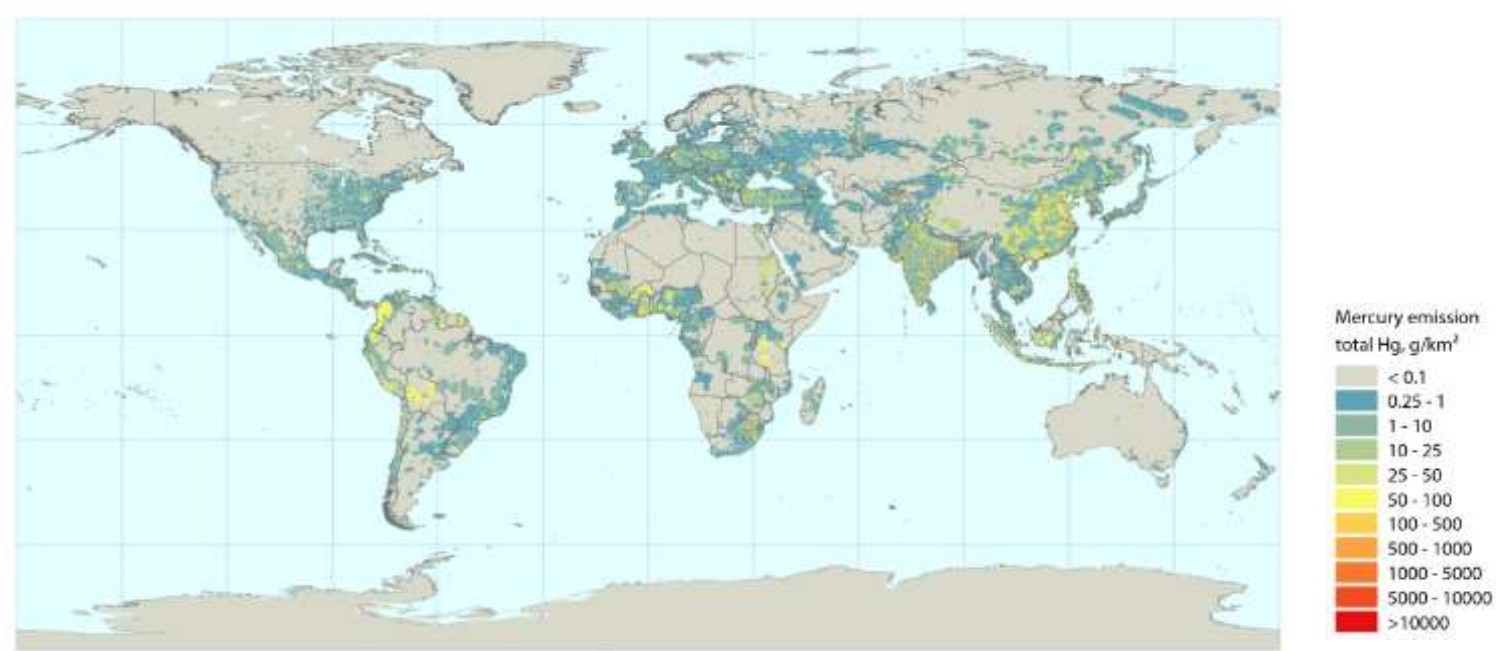

New policy

**Figure 4 b. Spatial distribution of mercury emissions in 2035 according the NP scenario.**

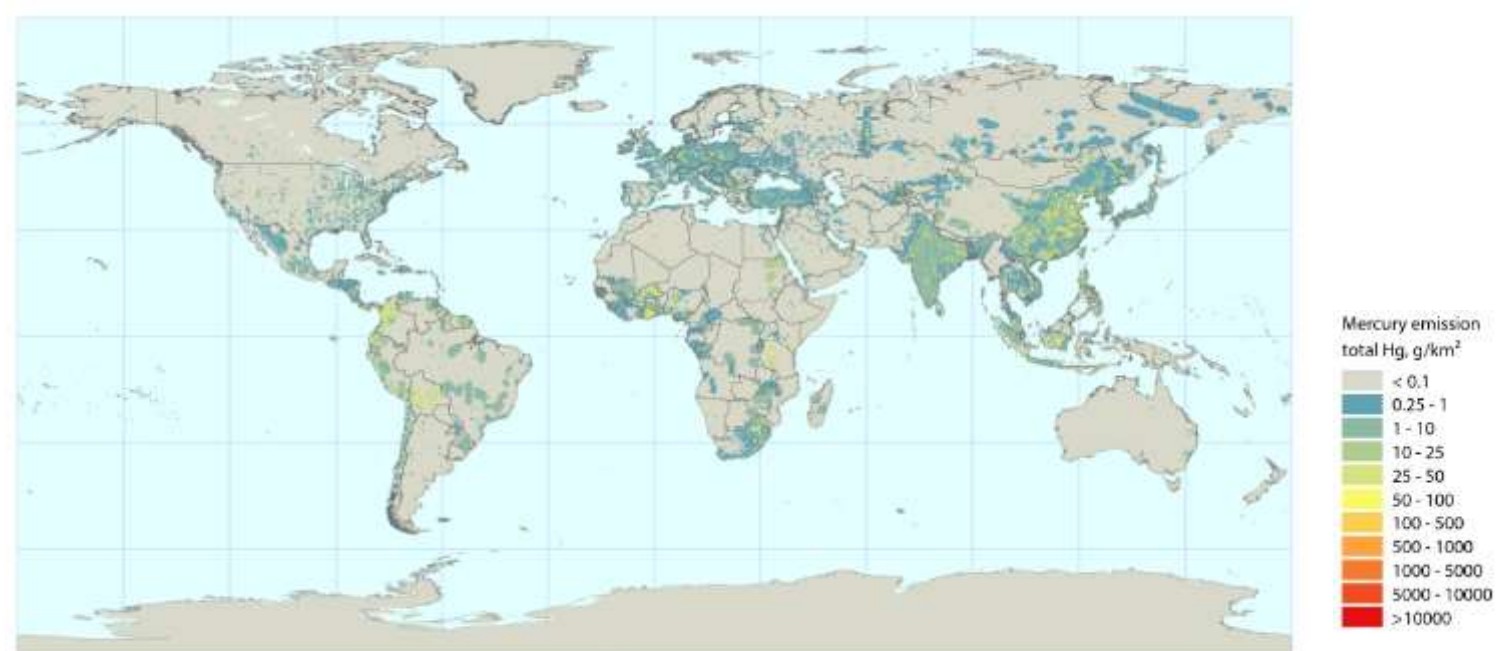

Maximum feasible reduction

**Figure 4 c. Spatial distribution of mercury emissions in 2035 according the MFR scenario.**

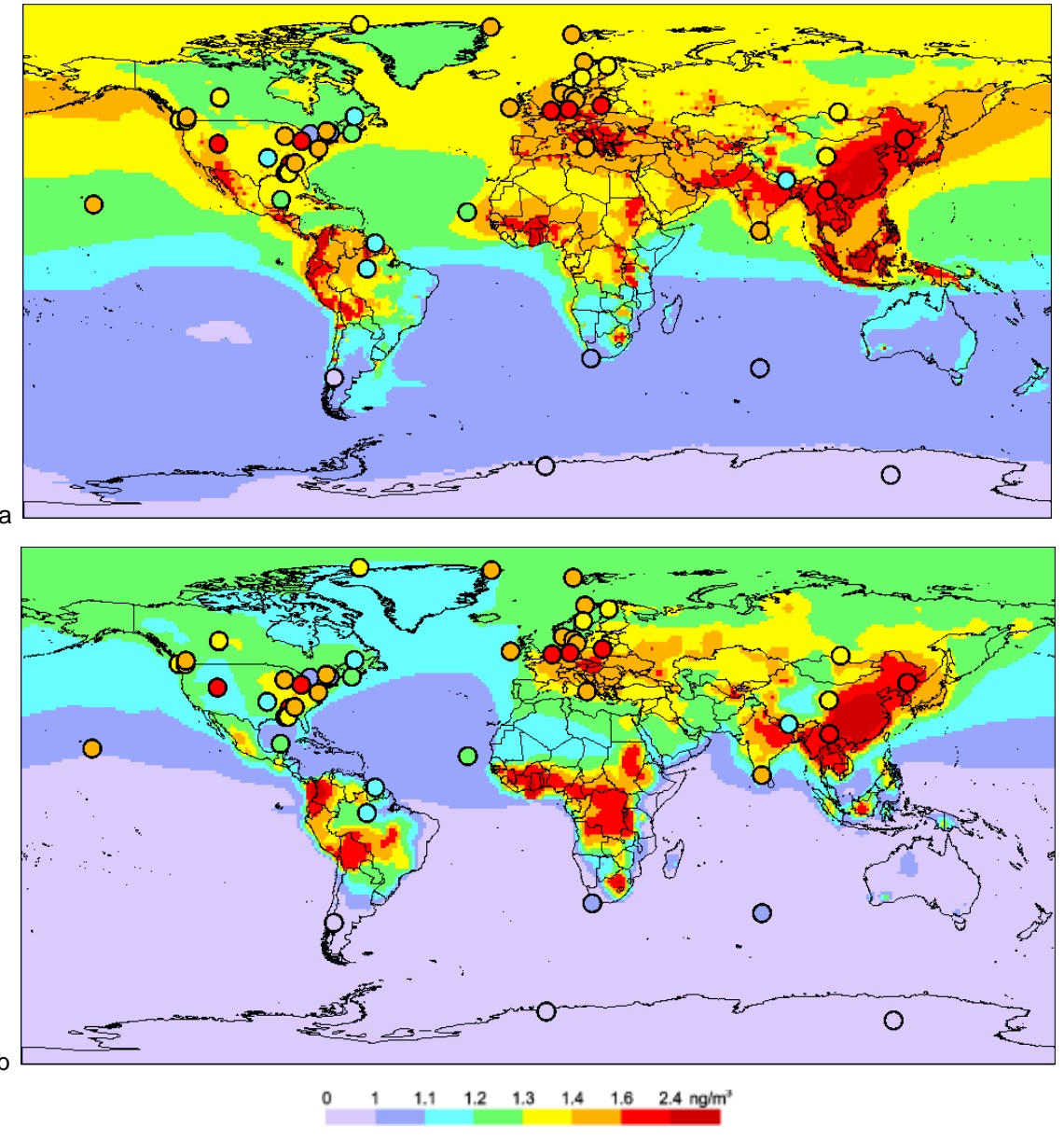

**Figure 5. Global distribution of annual mean GEM concentration in near-surface air in 2013 simulated by GLEMOS (a) and ECHMERIT (b). Circles present observed values at ground-based monitoring sites.**

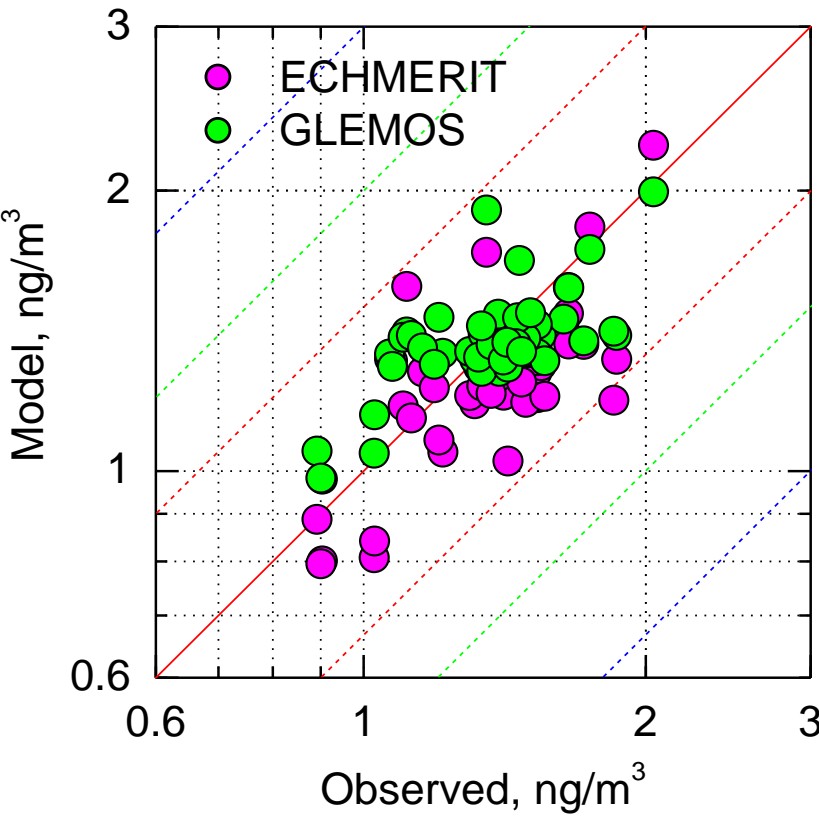

**Figure 6. Scatter-plots of annual mean GEM air concentration in 2013 measured at ground-based sites (Annex A, Table A.1) vs. simulated by GLEMOS and ECHMERIT. Red solid line depicts the 1:1 ratio; dashed lines show different deviation levels: red – by factor of 1.5, green – by factor of 2, blue – by factor of 3.**

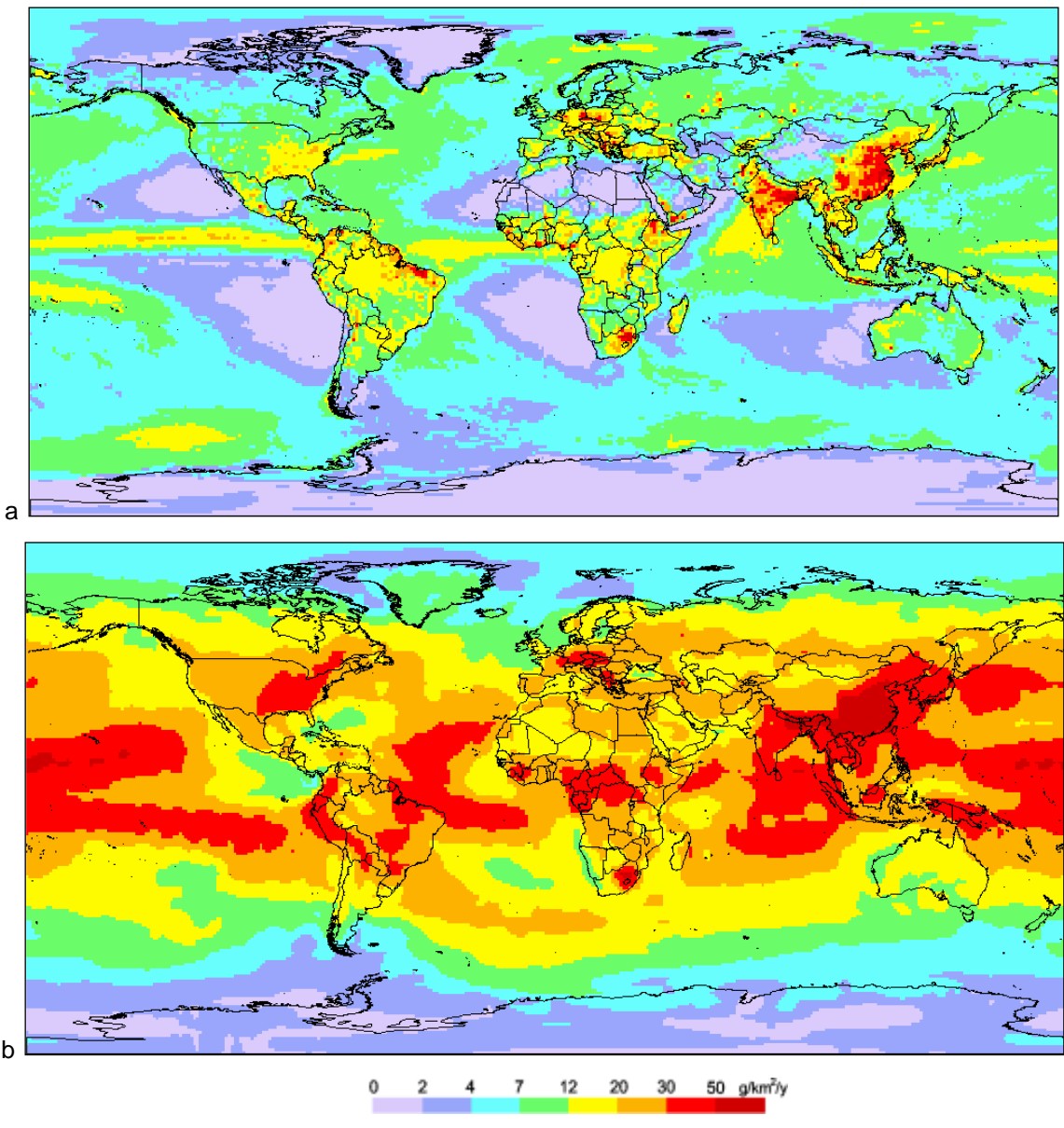

**Figure 7. Global distribution of total (wet+dry) mercury deposition flux in 2013 simulated by GLEMOS (a) and ECHMERIT (b).**

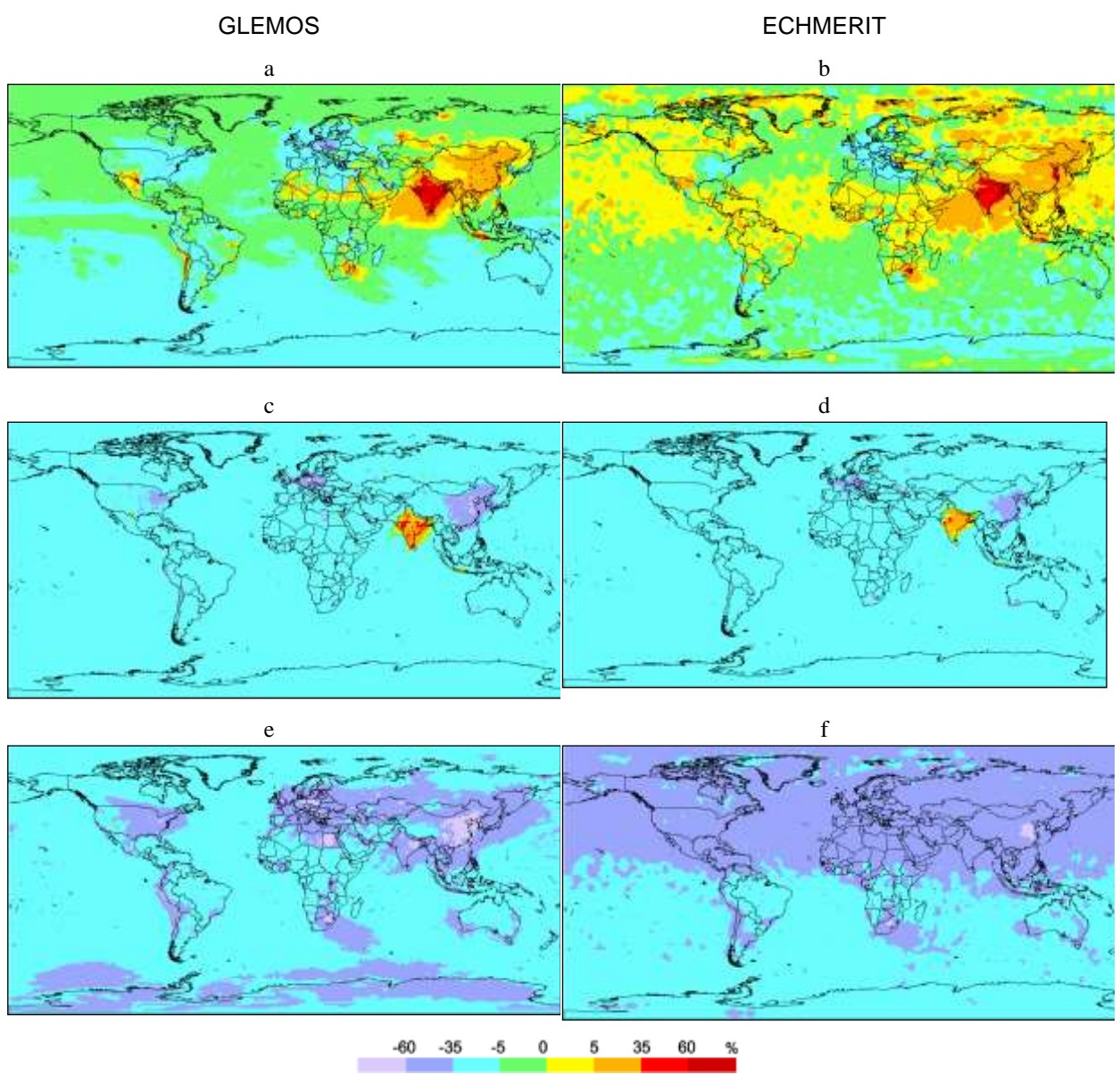

**Figure 8. Global distribution of relative changes of mercury deposition flux between 2010 and 2035 with respect to 3 emission scenarios: (a, b) – CP2035; (c, d) – NP2035; (e, f) – MFR2035. Results of GLEMOS and ECHMERIT simulations are presented in the left and right columns, respectively.**

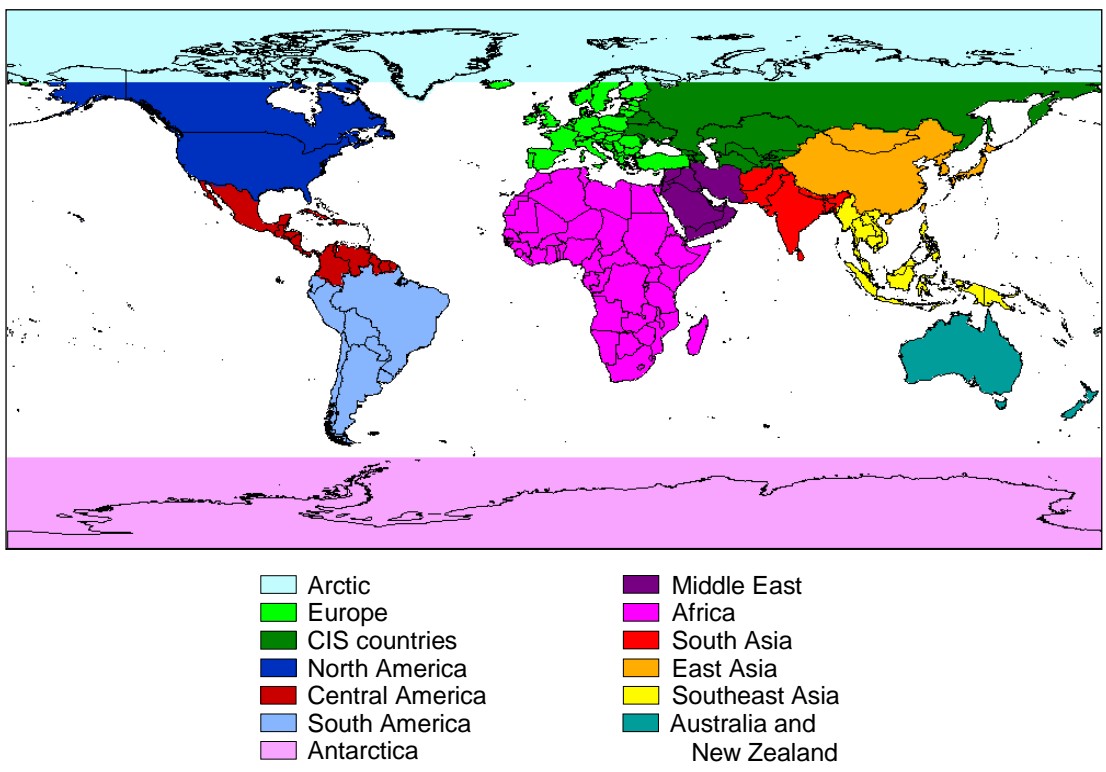

**Figure 9. Definition of source and receptor regions used in the analysis.**

Legend:
- Arctic
- Europe
- CIS countries
- North America
- Central America
- South America
- Antarctica
- Middle East
- Africa
- South Asia
- East Asia
- Southeast Asia
- Australia and New Zealand

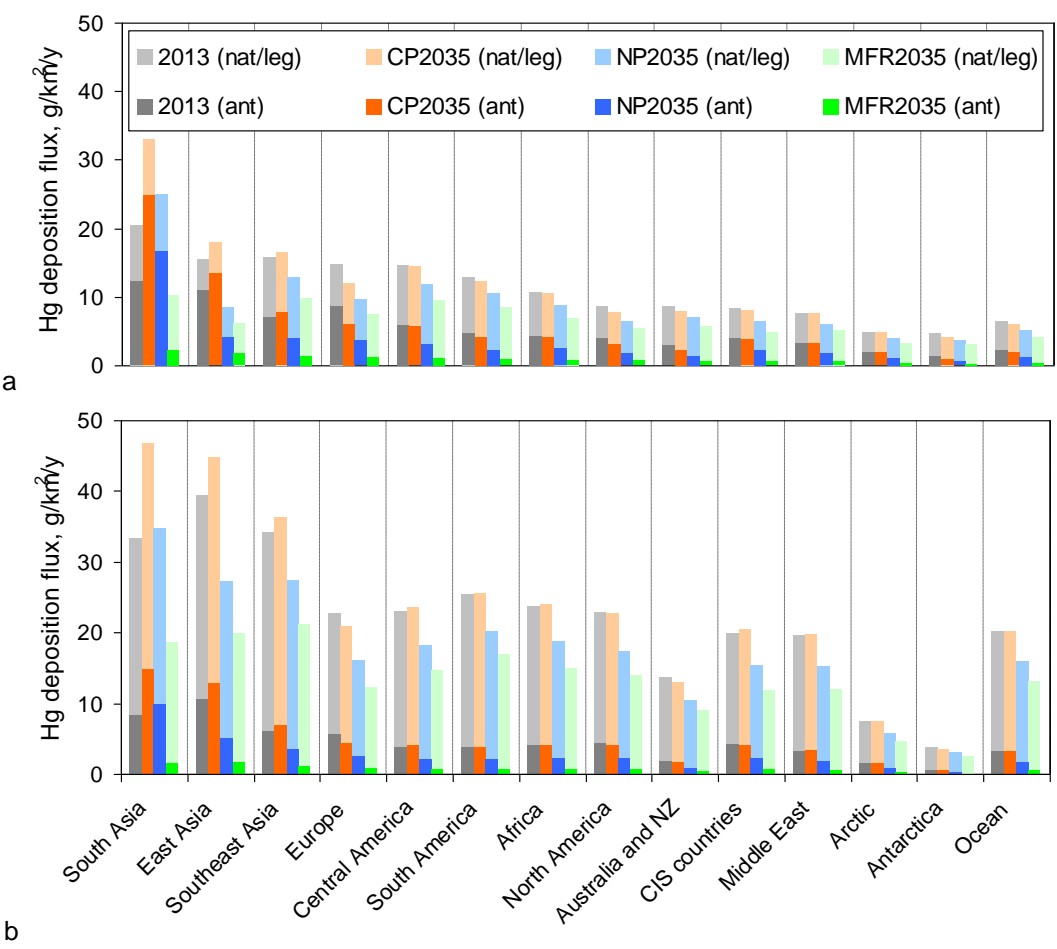

a

b

**Figure 10. Average mercury deposition flux in various geographical regions in 2013 and 2035 corresponding to the selected emission scenarios as simulated by GLEMOS (a) and ECHMERIT (b).**

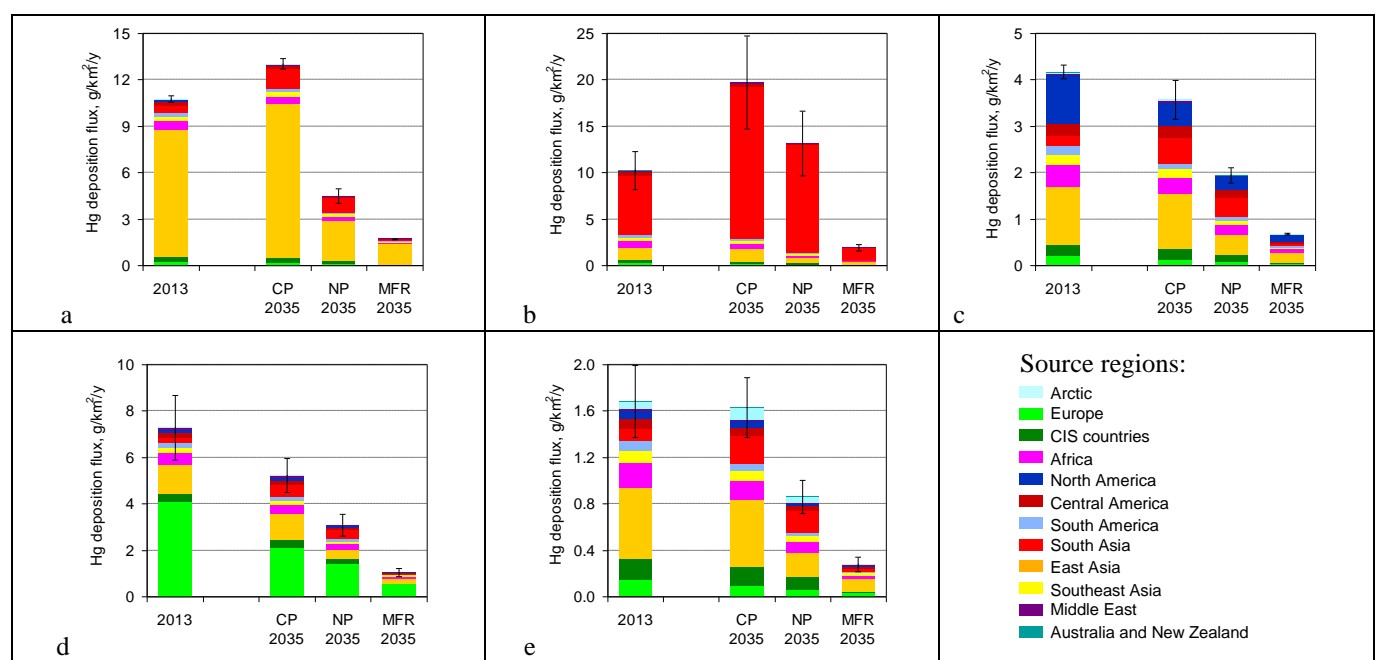

**Figure 11. Source apportionment of mercury deposition from direct anthropogenic sources (average of two models) in 2013 and 2035 in various geographical regions: (a) – East Asia; (b) – South America; (c) – North America a; (d) – Europe; (e) – Arctic. Whiskers show deviation between the models. Note different scales of the diagrams for different regions.**

