# Peer review of "Current and future levels of mercury atmospheric pollution on global scale"

_Atmospheric Chemistry and Physics, 2016_

## Referee Comment (RC1) · Anonymous Referee #1 · 30 Jun 2016

General Comments: This manuscript is the latest in a series of useful emissions inventories and modeling results produced through support of the Global Mercury Observation System (GMOS) project. The first part of the manuscript provides updated present and 2035 emissions estimates. Future scenarios consider a variety of uncertainties about trajectories of growth and implementation of air pollution control technology. The second part of the manuscript contains associated present and future deposition scenarios.

I have a few general comments that the authors can consider and try to incorporate into the manuscript text.

1) I found the information on how the emission inventories were constructed to be vague and lacking sufficient detail to enable reconstruction. This is a particular problem

right now as the authors point out because estimates across groups vary by a factor of 3. I wonder if the factor of three is overstated given convergence in estimates around the energy sector and then much larger uncertainties around artisanal and small scale gold mining. Given these uncertainties, presentation and availability of underlying data used to calculate emissions scenarios are essential and I think should be included in supplemental material accompanying this paper. The authors did not discuss other recent emissions estimates and compare/contrast with their own work and I think this would also be a helpful addition to the manuscript. For example, on page 4, the GDP per capita PPP and industrial goods relationships across countries.

2) I found the language on the GMOS project on page 2, first paragraph and concluding sections to be a distraction from the science presented in this paper. I think the authors should include text about the administration and goals/support of the project since this is not appropriate in this type of manuscript. I am supportive of mentioning GMOS as a successful endeavor but this does not seem like the right forum for the amount of text devoted to promoting it.

3) I don't think evasion of Hg0 from soils and the oceans should be referred to as "natural emissions" – this is propagating misinformation (and underestimating) the extent of human impact on the global Hg cycle. Other modeling studies suggest that these sources are now mainly reemitted legacy anthropogenic Hg. This is critical for understanding future mercury deposition scenarios as well and not mentioned in this paper. The actual present-day estimates for emissions from land and the ocean also do not seem to reflect our best present understanding of these sources and should be enhanced in the context of recent literature.

4) Related to point 3), the modeling approach presented assumes static reemissions from the land and oceans and this is a simplification of global biogeochemical cycle. It should be stated clearly as a limitation of the present analysis. See recent papers (Amos et al., 2013, 2014, 2015) for a thorough discussion of this and the implications of various uncertainties in the global Hg cycle.

5) The models applied here (GLEMOS and ECHMERIT) both use ozone and OH as the dominant global oxidants but this also seems out of date with our best present understanding. Br is mentioned but many additional oxidants are now on the table and a brief review of halogen chemistry and its implications for the global cycle and uncertainties in the context of this work also seem appropriate to include.

6) This manuscript is a bit light on citations – particularly the opening and closing discussions.

Specific editorial comments:

I would prefer to see the information on page 4 under "Database on activities" as a prose discussion rather than these bullets.

Page 4. Figure 1 does not seem very informative to me. Suggest adding to a supplemental section or deleting

Page 5 – since the ASGM emissions are such an important uncertainty, I would particularly like to see detailed information on how emissions estimates were constructed.

Page 9. Can Figures 5-7 be condensed at all?

Page 10. "prompt re-emission (Selin et al., 2008)" is a non-physical process intended to help tune the model. I don't think this should be a part of other models. The authors might instead consider the uncertainty around photoreduction and evasion, particularly over ice and snow covered surfaces.

Page 10. See general comments on oxidants. Shah et al., 2016, ACP is relevant here.

Page 10. The air-sea exchange scheme in DeSimone et al., 2014 is based on prescribed uniform ocean concentrations of elemental Hg so thus does not consider variability in ocean concentrations and changes in ocean concentrations as a function of time. This should be acknowledged.

Page 11. Discussion on terrestrial emissions seems out of date to me as well. Consider

synthesizing and incorporating information from recent diverse terrestrial Hg cycling literature and also global modeling (i.e., Smith-Downey et al., 2010).

Page 13. Future deposition scenarios should acknowledge static global reservoirs and also I assume the same meteorological years for future deposition? I think addressing biogeochemical variability is beyond the scope of this work but it would be good to acknowledge as an uncertainty – particularly given the rapidly changing climate.

Page 14. My impression was that volcanoes are actually pretty well constrained as a source. See Amos et al. (2015).
* * *

---

## Referee Comment (RC2) · Anonymous Referee #2 · 5 Jul 2016

In general, this is a thorough presentation of a new inventory for mercury and a test of its application in global modeling. It is a rather straightforward analysis, appropriately conducted. However, the paper doesn't really do enough analysis to identify new information of relevance to global mercury science. I would suggest that the authors revise to more strongly identify either 1) the differences between this and previous inventories, or 2) the ability of modeling to constrain what we know about anthropogenic mercury. I would also suggest that the authors be more specific about the scientific questions they are trying to answer.

Specific comments follow:

abstract: one can't 'assess' future concentrations. Rephrase here and elsewhere throughout the draft.

[Figure]

page 1 line 26: re-emissions aren't "natural processes" — suggest using another word

page 2 line 1-5: this information seems like PR for the GMOS project and I think it is out of place in a scientific publication.

Page 2 line 10: "This contaminant does not degrade easily in the environment": Mercury, as an element, does not degrade *at all* in the environment. Rephrase.

Page 2, line 16: Ambio, 2007 is an incorrect citation

Page 2, lines 20-25: In a paper focusing on global mercury pollution, this introductory focus on activities in the EU seems misplaced.

Page 3, lines 9-13: These questions are all important, but none of them is convincingly answered in the paper (and in fact are too ambitious for any one paper). It would be helpful if the authors narrowed this scope a bit.

page 3, lines 17-20: I get that this study is EU funded, but this information really should only be in an acknowledgment.

page 4, line 4-6: Is the goal for this inventory to be comparable with AMAP/UNEP 2013 or is the method just consistent? This should be clarified, as previous AMAP/UNEP emissions inventories are not directly comparable.

page 8, line 18: IPCC acronym is incorrect.

page 8 line 22: this is the first reference to these acronyms (CP, NP, MFR) and they need to be explained

Section 2.5, 2.6: much more detail here is needed on the assumptions of these scenarios. In particular, the description of the 2035 scenario is not well described — it is unclear even which energy scenario was used for it, for example.

p 9 line 31: how is biomass burning treated in the new inventory?

p 10: I would argue that the current evidence for the Br mechanism is stronger than the authors give credit to. However, from a global budget perspective, the choice of mechanism doesn't affect the question and results. I would suggest the authors focus on this point rather than try to defend an outdated mechanism.

page 10-11: If ECHMERIT has an online parameterization of natural and secondary emissions, shouldn't the results include at least the (shorter timescale) response of surface reservoirs? Clarify.

page 15: line 17-28: This text on GMOS is again over-the-top and not appropriate in a scientific paper.

Annex a: It would be useful to list quantitative mean values for measured and simulated Hg at these sites. Not all of these results have been published in the peer-reviewed literature, as far as I can tell

---

## Author Response (AR1)

**Response to comments – J. Pacyna**

**Reviewer No. 1.**

Comment 1.

More information on how the emission inventories were constructed has been added to the support material (Annex A).

Comment 2.

Agree. The changes were introduced to the manuscript.

Comment 3.

Agree. Indeed, the evasion of Hg from soils and oceans seem to be more re-emitted legacy anthropogenic sources rather than natural sources.

Appropriate changes were introduced to the manuscript.

Comment 4

An appropriate discussion included.

Comment 5

Detailed discussion of the chemistry is beyond the scope of the paper. Nevertheless, the text on applied chemical schemes has been revised in line with recommendation of the reviewer.

Comment 6

New literature sources added.

Specific editorial comments

Page 4:

Fig. 1 moved to Annex A

Page 5:

Information added.

Page 9:

Done

Page 10:

The description of model parameterization of re-emission processes, atmospheric chemistry and air-water exchanged ahs been revised.

Page 11:

The text and the references have been updated.

Page 13:

Appropriate discussion added

Page 14:

Corrected

**Reviewer 2**

Page 1:

Done

Page 2: line 1-5

This chapter was removed.

Page 2 line 10

Corrected

Page 2, line 16;

I disagree, Ambio (2007) is the right citation. Please, see this special edition of the journal.

Page 2, line 9-13

I disagree with the comment. There is no focus on EU in the introductory. This is simply putting the paper in context of a need for information about sources, emission, and transport of mercury in order to take decision on reduce emissions and exposure to this element, as outlined in the EU Mercury Strategy and the Minamata Convention.

Page 3, lines 9 – 13

I agree, the revision was made.

Page 3, lines 17 – 20

A slight revision was made.

Page 4, line 4 -6:

The methodology for estimating future Hg emissions is consistent with the methodology developed in AMAP/ UNEP (2013) The authors of the manuscript are also the authors of the AMAP/ UNEP (2013) report. Some adjustment is made in the text.

Page 8, line 8

Corrected.

Page 8, line 22:

Done.

Section 2.5 and 2.6:

The assumptions of the scenarios are described separately in section 2.4 – Definition of emission scenarios. A more detailed description is added to section 2.6. An illustration of the assumed coal use until 2035 under the WEO CP (Current Policy), NP (New Policy) and 450 scenario assumptions has been added to the support material (Annex A). A reference to this illustration is provided in section 2.6.

Page 10:

Agree. The text on the atmospheric chemistry has been completely re-written.

Page 10-11:

The model parameterization of the air-seawater exchange is based on static values of Hg in seawater so it cannot reflect response of the ocean. Appropriate remarks added.

Page 15, line 17-28:

I agree. The text about GMOS is removed.

Annex a:

This Annex will be withdrawn as it does not relate directly to the work presented in the paper.

**Current and future levels of mercury atmospheric pollution on global scale**

**Jozef M. Pacyna** [a,f], **Oleg Travnikov** [b], **Francesco De Simone** [c], **Ian M. Hedgecock** [c],

**Kyrre Sundseth** [a], **Elisabeth G. Pacyna** [a], **Frits Steenhuisen** [d], **Nicola Pirrone** [e],

**John Munthe** [g] **and Karin Kindbom** [g]

[revised manuscript text omitted]